# Inborn Errors of Immunity Predisposing to Herpes Simplex Virus Infections of the Central Nervous System

**DOI:** 10.3390/pathogens12020310

**Published:** 2023-02-13

**Authors:** Morten Kelder Skouboe, Marvin Werner, Trine H. Mogensen

**Affiliations:** 1Department of Infectious Diseases, Aarhus University Hospital, 8000 Aarhus, Denmark; 2Department of Biomedicine, Aarhus University, 8000 Aarhus, Denmark

**Keywords:** herpes, HSV, encephalitis, meningitis, interferon, genetic defects, TLR3

## Abstract

Herpesvirus infections can lead to a number of severe clinical manifestations, particularly when involving the central nervous system (CNS), causing encephalitis and meningitis. However, understanding of the host factors conferring increased susceptibility to these diseases and their complications remains incomplete. Previous studies have uncovered defects in the innate Toll-like receptor 3 pathway and production of type I interferon (IFN-I) in children and adults that predispose them to herpes simplex encephalitis. More recently, there is accumulating evidence for an important role of IFN-independent cell-autonomous intrinsic mechanisms, including small nucleolar RNAs, RNA lariat metabolism, and autophagy, in restricting herpesvirus replication and conferring protection against CNS infection. The present review first describes clinical manifestations of HSV infection with a focus on neurological complications and then summarizes the host–pathogen interactions and innate immune pathways responsible for sensing herpesviruses and triggering antiviral responses and immunity. Next, we review the current landscape of inborn errors of immunity and the underlying genetic defects and disturbances of cellular immune pathways that confer increased susceptibility to HSV infection in CNS. Ultimately, we discuss some of the present outstanding unanswered questions relating to inborn errors of immunity and HSV CNS infection together with some perspectives and future directions for research in the pathogenesis of these severe diseases in humans.

## 1. Herpes Simplex Virus Infections of the Central Nervous System

Herpes simplex virus types 1 and 2 (HSV-1/-2) are ancient human pathogens with a wide spectrum of disease manifestations. These double-stranded (ds)DNA viruses have been remarkably successful from an evolutionary point of view: computational models have shown that HSV-1 is likely the result of codivergence from chimpanzee herpesvirus around 6 Ma, while HSV-2 has arisen from a cross-species transmission event of the same virus 1.6 Ma [1]. It should, therefore, come as no surprise that both HSVs are prevalent in all parts of the world with an estimated two thirds of the global population being infected with at least one of these viruses [2]. The key to understanding this success lies in the ability of both viruses to have two distinct replication cycles: lytic and latent.

Upon primary infection, HSV will start its lytic replication cycle in epithelial cells, causing cold sores (mostly HSV-1) or genital sores (HSV-2). Eventually, the virus will invade sensory neurons, traveling retrogradely to the soma and establishing latency, particularly in the trigeminal and sacral ganglia for HSV-1 and HSV-2, respectively. From here, the virus can occasionally reactivate and travel anterogradely to the primary infection site, causing recurrence of the sores or genital herpes and enabling transmission to the next host. However, on rare occasions, HSV will instead invade the central nervous system (CNS), either during primary infection or at reactivation, thereby giving rise to herpes simplex encephalitis (HSE), brainstem encephalitis, or aseptic meningitis [3]. 

HSE is the most common form of sporadic viral encephalitis, accounting for 10–20% of all cases, with HSV-1 causing >90% of HSE [4]. There are two age peaks of the disease, one between 5 and 30 years of age and one over 50 years of age [4]. The incidence of HSE is estimated to be around 1–2/250,000 per year, with neither geographical nor seasonal bias [5,6]. Patients usually present with fever, confusion, decreased consciousness, and nuchal rigidity [4,7], and the diagnosis is made by detection of pleocytosis and HSV DNA in the cerebrospinal fluid [8]. Before effective antiviral treatment was developed in the form of acyclovir, mortality approached 70%, and even with adequate antiviral therapy, it can still be as high as 25%, with the majority of survivors suffering from temporary or permanent neurological sequelae, including anterograde amnesia and cognitive deficits [9]. 

Brainstem encephalitis, also called rhombencephalitis, caused by HSV is extremely rare with only a few dozen published cases. Patients from case reports have been described as having a fever and varying neurological symptoms, including abnormal ocular movements, cranial nerve deficits, and headaches [10]. Acyclovir treatment appears to be beneficial, despite the low number of patients described [10]. 

Aseptic meningitis is defined by inflammation of the meninges without bacterial infection. In most cases, the exact etiology cannot be determined [11,12]. In the cases caused by viruses, enteroviruses, varicella zoster virus, and HSV-2 are the most common agents [6,12]. The most common symptoms include headache, nausea, nuchal rigidity, and fever [6]. Mortality is generally low, but a significant proportion of patients have unfavorable outcomes nonetheless [6]. In rare cases, patients suffer from recurrent episodes of viral meningitis, known as Mollaret’s meningitis, named after the French neurologist Pierre Mollaret, who first described this condition [13]. The most common etiology in these patients is HSV-2, and the condition is usually self-limiting, although half of patients experience transient neurological impairment [14]. Acyclovir and valacyclovir are thought of as effective, although definitive trial data are for antivirals as prophylaxis and treatment are lacking [14]. 

It is this conundrum of the ubiquity of HSV in the global population and its rare but serious CNS manifestations that has spurred research for decades to understand the interplay between HSV and host immunity. 

## 2. Immunity to Herpes Simplex Virus

HSV has been successfully transmitting and adapting to human immunity for millions of years, while still retaining its virulence. So far, 284 viral open reading frames (ORFs) have been identified, the transcripts and protein products of which modulate and help the virus evade the adaptive and innate immune responses as well as cell-intrinsic defenses [15,16,17]. As HSV-1 and HSV-2 share 83% homology in their genes, and several of their proteins share biological functions [18], we will for, the most part, not discriminate between the two viruses in the following sections.

### 2.1. Interferon-Dependent Anti-HSV Immunity

Secretion of type I interferons (IFN-I) is a central part of anti-HSV immunity (Figure 1). In the extracellular and endosomal compartments, HSV is recognized by at least three Toll-like receptors (TLR), all of which eventually induce the expression of IFN-I. TLR3 is activated by dsRNA species to signal through interferon regulatory factor (IRF)3, IRF7, or nuclear factor (NF)-kB [19,20]. TLR2 recognizes HSV glycoprotein B (gB) and signals through myeloid differentiation primary response (MyD)88 and NF-κB [21]. TLR9 recognizes DNA with unmethylated CpG motifs, such as the HSV genome, and signals through either MyD88 and NF-κB or via MyD88 and IRF7 [22,23].

In the cytosol, multiple receptors exist to detect viral DNA and other danger signals, such as dsRNA species. Notable here is the cyclic GMP-AMP synthase–stimulator of interferon genes (cGAS-STING) axis. cGAS recognizes dsDNA from the viral genome or mitochondrial DNA leaked into the cytosol [24] to produce 2′3′-cyclic GMP-AMP (2′3′-cGAMP) as a second messenger to activate STING on the endoplasmic reticulum [25,26,27]. STING then signals through serine/threonine protein kinase 1 (formerly known as TANK-binding kinase 1, TBK1) and IRF3 to induce IFN-I [16]. Alternative dsDNA sensors exist, including DNA-dependent activators of IRFs (DAI) [28] and RNA polymerase III (POL III) [29,30], but the cGAS-STING signaling axis has been recognized as a major contributor to dsDNA-induced antiviral signaling, at least in mouse studies [31,32], although this is likely tissue and cell-type dependent. The presence of dsRNA in the cytosol produced during viral replication by POL III recognizing dsDNA [29,30] or through combined upregulation and increased nuclear export of 5S rRNA 141 (*RNA5SP141*) [33] will activate retinoic acid inducible gene (RIG)-I (and, to a lesser degree, melanoma differentiation-associated gene (MDA)5) and start signaling through mitochondrial antiviral-signaling protein (MAVS), TBK1, and IRF3 to induce IFN-I [34,35,36,37]. Alternatively, dsRNA can stimulate protein kinase R (PKR), which, through phosphorylation of eukaryotic initiation factor (eIF)2α, will halt translation of host and viral protein alike [38], or it can activate the oligoadenylate synthetase (OAS)-RNase L system which degrades RNA in the cytosol and inhibits viral replication [39]. Finally, small self-RNA products from RNase L activity can also serve as RIG-I ligands, further amplifying the antiviral response [40]. 

The above-mentioned pathways will eventually lead to the upregulation of IFN-I which includes IFN-α (13 subtypes), IFN-β, IFN-ε, IFN-κ, and IFN-ω. Two other classes of IFN exist (IFN-II and IFN-III), but these have only been implicated in defects in anti-HSV immunity when the IFN-I system is also affected due to overlap in the downstream signaling cascades, and they will, therefore, only be mentioned briefly. IFN-I are broad-acting antiviral cytokines that signal through the IFN-α/β receptor (IFNAR), composed of IFNAR1 and IFNAR2. IFNAR recruits Janus kinase (JAK)1 and tyrosine kinase (TYK)2, which, in turn, phosphorylate the signal transducer and activator of transcription (STAT)1 and STAT2, which then, together with IRF9, form the complex IFN-stimulated gene factor (ISGF)3 to induce hundreds of IFN-stimulated genes (ISGs). These ISGs are the main drivers of the antiviral response and include PKR, RNase L, and RIG-I [41]. In the periphery, IFN-β can be produced by almost all cells, whereas specialized leukocytes, plasmacytoid dendritic cells (pDC), are the main IFN-α factories [42,43]. The source of IFN-I in the CNS is less obvious. Almost all CNS cells can produce IFN-I, but an in vivo study has suggested microglia and astrocytes are the primary IFN-I producers upon infection with HSV-1 and that neurons play a minor role in the production of IFN-I upon HSV-1 infection [44]. 

The importance of the above pathways is perhaps best illustrated by the many immune evasion strategies employed by HSV. For example, infected cell protein (ICP)0 inhibits nuclear translocation of IRF3, which is central to several pathways [45]. Viral protein (VP)1-2 and ICP27 both inhibit STING [46,47], and likewise ICP27 also inhibits STAT1-mediated signaling [48]. The function of PKR is also targeted by Us11, sequestering dsRNA and binding directly to PKR [49,50], as well as by ICP34.5 recruiting protein phosphatase 1α to disinhibit protein translation [51]. 

A simplified overview of these IFN-I dependent pathways is provided in Figure 1.

### 2.2. Interferon-Independent Anti-HSV Immunity

#### 2.2.1. Autophagy

Autophagy is an essential mechanism for the cell to dispose of intracellular components by lysosomal degradation. It requires a set of autophagy-related proteins (ATG) and is subdivided based on delivery mechanism into macro-autophagy, micro-autophagy, and chaperone-mediated autophagy, of which macro-autophagy is the best characterized process. Macro-autophagy (hereafter simply ‘autophagy’) is triggered by many different stimuli, including starvation, cellular stress, and infection. Structures can be targeted for degradation by ubiquitination which is recognized by p62 (also known as sequestosome 1 or SQSTM1) or other autophagy receptors. The cargo is engulfed by a double-membrane vesicle termed the autophagosome. Microtubule-associated protein 1 light chain 3 (LC3) plays a key role in the formation and maturation of the autophagosome, and lipidated LC3 (LC3-II) accumulates when autophagy is activated, or lysosomal clearance is blocked [52]. Autophagy plays both anti- and pro-viral roles during the HSV life cycle, and the net effects are often cell-type or tissue-dependent. HSV-1 particles are suggested to be degraded by autophagy [53], and autophagy can facilitate the loading of HSV-1 gB onto MHC-I molecules [54]. Control of HSV-1 in primary neurons is critically dependent on ATG5, whereas this is not the case for mucosal epithelial cells [55]. HSV-1 also encodes two anti-autophagy proteins: ICP34.5 inhibits Beclin-1 which is another essential part of autophagosome maturation [56]; and Us11 inhibits PKR and its ability to stimulate autophagy [57]. Although HSV-2 also seems to modulate autophagy, no viral autophagy-blocking proteins have been identified so far [52], so different roles for autophagy between the two viruses cannot be excluded. Finally, an siRNA-knockdown screen has revealed differential roles of autophagy-related proteins in HSV-1 replication [58]. 

#### 2.2.2. The Lectin Pathway

The lectin pathway of the complement system recognizes surfaces displaying an array of carbohydrates or acetyl groups. The pattern recognition molecules (collectively referred to as lectins) include mannan-binding lectin (MBL), ficolins 1-3, and collectin 10 and 11. The lectins circulate in complex with MBL-associated serine proteases (MASPs) which activate the rest of the pathway [59]. Both MASP1 and MASP2 are indispensable for activation of the lectin pathway [60], and the lectin pathway has been shown to modulate the immune response to HSV infection in vivo through recognition of viral glycoproteins [61]. HSV counters the complement pathways broadly by expression of its surface protein gC, which blocks C3-activity [62]. 

## 3. Inborn Errors of Interferon-Dependent Innate Immunity

As previously stated, IFN-I is central to antiviral immunity. IEI in the pathways activating IFN-I or the IFN-I signaling axis itself have been well documented to cause increased susceptibility to a range of viruses, most notably live-attenuated viral vaccines (LAV), respiratory viruses, and herpesviruses [63]. Specific genetic variants previously described in connection with HSV CNS infections are compiled in Table 1.

### 3.1. TLR3 Signaling

The TLR3 signaling axis has emerged as a non-redundant and essential part of the defense against HSE. In general, cells from patients with these defects show decreased IFN-I responses to HSV-1 and increased cytolytic activity or production of the virus, even impacting the intrinsic defenses of induced pluripotent stem cell (iPSC)-derived neurons [84]. *UNC93B1* deficiency in two unrelated patients with HSE was reported to reduce signaling via TLR3 and TLR7-9 due to its role in trafficking these receptors from the ER to the endosomes [64]. Both autosomal recessive (AR) and dominant (AD) defects of *TLR3* itself have been reported and exhibit specific susceptibility to HSV-1 [65,66,67]. Defects in downstream signaling adaptors have also been documented, including *TRIF*, *TRAF3*, *NEMO*, *TBK1*, and *IRF3* (Figure 1 and Table 1) [68,69,70,71,72,73,74]. Although other signaling cascades converge on some of these downstream adaptors, including STING and RIG-I-MAVS signaling, TLR3 signaling appears to be the common denominator for all these defects, underscoring its importance in HSV-1 immunity of the CNS. A possible explanation for the central role of TLR3 is its recently proposed function as a rheostat of tonic IFNAR signaling, as discussed below [85]. 

### 3.2. IFNAR Signaling

Defects in the IFNAR signaling cascade can have a spectrum of clinical pictures, including severe disseminated infection after LAV administration or during natural viral infections, and is mostly determined by the localization of the specific defect within the signaling cascade [63]. This is partly due to several adaptor proteins, especially in the JAK-STAT signaling, being shared between the receptors for IFN-I and IFN-II. 

*IFNAR1* deficiency, initially associated with severe disease after LAV [86], was recently linked to a case of childhood HSE [76]. The identified variant encoded a truncated IFNAR1 protein which could not interact with TYK2. This is the first time HSE has been directly attributed to IFNAR deficiency [87]. The explanation could be a broader phenotype for this immunodeficiency, and patients, therefore, usually present with some other severe infectious phenotype, usually heightened susceptibility to LAV [63,86,88,89], as well as a short observation time (the first case of IFNAR deficiency was only described in 2015 [63,88]).

Deficiency of *STAT1* was the first reported genetic etiology of HSE [77,90]. The patient had dissemination of the Bacillus Calmette–Guerin mycobacterial vaccine and recurrent HSE which proved lethal in the second episode [77]. This patient was included in a recent study describing a cohort of 32 patients with complete or partial *STAT1* deficiency, and it was the only case of HSE, although several other patients had experienced mild HSV-related disease [91]. A related IEI, *TYK2* deficiency, is also linked to disseminated mycobacterial disease. At least one patient has been reported with HSV meningitis, two others with aseptic meningitis of unknown etiology, and one patient with HSE [82,92,93]. 

Deficiency of *IRF9* due to a biallelic splice-site variant has been reported in a family with heightened susceptibility to a range of viruses and sepsis [78]. One sibling was admitted to the pediatric intensive care unit for HSV meningoencephalitis at the age of 9 months. At 14 months of age, she died due to complications following a yellow fever virus vaccination [78], mirroring the phenotype of IFNAR deficiency with heightened susceptibility to LAV [63,86,89].

### 3.3. GTF3A and RNA5SP141 as Guard Mechanisms

Recently, Naesens and colleagues reported a case of HSE in a patient who was compound heterozygous for variants in *GTF3A* [75]. This gene encodes the transcription factor IIIA (TFIIIA), which is essential for POL III-mediated transcription of 5S rRNA species. Among these is the pseudogene *RNA5SP141*, which was previously shown by the Gack group to play an essential role in antiviral immunity [33]. The RNA species (5S rRNA 141) is normally expressed at low levels and mainly resides in the nucleus, and the few copies that make it to the cytoplasm are sequestered by the RNA-binding proteins MRPL18 and TST. HSV-infection, through its virion host shutoff (vhs) protein, downregulates most host mRNA, including MRPL18 and TST, but *RNA5SP141* is slightly upregulated, and most of it is then exported to the cytoplasm [33]. In a mechanism analogous to the guard theory of plant immunity [94], RNA 5S ribosomal 141 is released from its sequestration and can now induce the RIG-I-mediated activation of IFN-I [33] (Figure 2A,B). Interestingly, both the reported patient and his sister (who also harbored both *GTF3A* variants but did not experience HSE) had decreased IFN-I responses to HSV-1, and this could be rescued by reconstitution with WT *GTF3A* [75]. Both siblings also had a CVID-like phenotype [75], and, although rare, HSE is not unheard of among CVID patients, as discussed below [95]. It remains to be clarified if the biallelic *GTF3A* variants disturb B cell function as well and whether the HSE phenotype manifests exclusively in the context of disturbed B cell function (primary or secondary).

## 4. Inborn Errors of Interferon-Independent Innate and Cell-Intrinsic Immunity

### 4.1. Autophagy

Building on the evidence of a neuron-specific antiviral role of autophagy in murine neurons [55], two cases of recurrent lymphocytic Mollaret’s meningitis caused by HSV-2 have been reported due to monoallelic variants in the essential autophagy genes *ATG4A* and *MAP1LC3B2*. Primary fibroblasts from both patients showed markedly decreased autophagy induction upon HSV-2 infection, starvation, or rapamycin treatment, as well as increased HSV-2 replication and cell death, all of which was rescued by reconstitution with WT *ATG4A* and *MAP1LC3B2* [83]. This study is supported by another study demonstrating that the HSV-1 protein ICP34.1 confers neurovirulence by targeting the autophagy protein Beclin-1 for degradation [56]. Altogether it may make sense that neurons, being post-mitotic cells sensitive to hyperinflammation and hypoxia, utilize autophagy in antiviral defenses to restrict excessive IFN-I production and immunopathology [96]. As mentioned above, not much is known about the role of autophagy in the HSV-2 life cycle and antiviral immunity because most HSV research has been carried out using HSV-1. It will, therefore, be interesting to learn if autophagy plays a similar role in neuroprotection against HSV-1 CNS infections in future studies. 

### 4.2. Intrinsic Defenses: SNORA31 and DBR1

In recent years, two studies from the Casanova group have uncovered defects in cell-intrinsic defense systems predisposing to brainstem encephalitis and HSE. 

In seven patients from three kindreds with brainstem encephalitis due to either HSV-1, influenza B virus, or norovirus, the group identified biallelic variants in the gene encoding debranching enzyme 1, *DBR1* [81], which hydrolyzes intron lariat RNA at its branch point to facilitate its turnover or processing. This enzyme is ubiquitously expressed, but the levels are highest in the spinal cord and brainstem. The group found that the variants caused a defect in DBR1 function due to an enzymatic defect as well as lower protein levels. Patient cells showed an increased HSV-1 replication rate and increased levels of RNA lariats. Together with the fact that HSV-1 has been shown to inhibit splicing or cause alternative splicing of some genes [97], the authors speculate that DBR1 controls brainstem-intrinsic defenses by regulation of lariat metabolism [81]. 

In five patients with HSE, the group found monoallelic missense mutations in *SNORA31* [79]. The gene encodes a small nucleolar RNA, snoRNA31. Fibroblasts and iPSC-derived neurons showed heightened susceptibility to HSV-1, whereas this was not observed in iPSC or B cells, suggesting a cell-type specific role for snoRNA31. The authors demonstrated that snoRNA31 is involved in pseudo-uridylation at U218 of 18S rRNA; however, this was only affected in full knockouts and not in cells carrying monoallelic variants, the latter being the case for all patients. Furthermore, the transcriptome was affected differently in patient cells infected with HSV-1 than in healthy control cells, but the exact mechanism by which snoRNA31 protects against HSV-1 remains elusive [79].

### 4.3. Lectin Pathway of Complement Activation

IEI of the lectin pathway have been shown to be associated with symptomatic HSV reactivation [61,98]. Bibert and colleagues reported two cases of HSE in adult patients who harbored monoallelic loss-of-function variants in *MASP2*. Their assays showed that antibody-depleted plasma from patients displayed reduced capacity for HSV-1 neutralization [80]. *MASP2* deficiency was first described in 2003 in a patient with severe pneumococcal infections as well as autoimmune and autoinflammatory conditions [99]. Since then, at least ten additional patients with various clinical pictures as well as nine asymptomatic individuals have been identified, casting doubt on the clinical relevance of the lectin pathway in human immunity [59]. MBL deficiency is the prime example of a related controversial condition, originally described as an immunodeficiency with susceptibility to pyogenic infections, albeit with low clinical penetrance [100], only to later have this status revoked due to the high frequency of asymptomatic MBL-deficient individuals [101,102,103]. However, several other proteins act in parallel with MBL, whereas MASP1 and MASP2 are both non-redundant in the lectin pathway, and the report of HSE in MASP2-deficient patients can, therefore, not be discarded on the same grounds. It should be noted, however, that systematic investigations of very large groups of individuals with IEI affecting the entire lectin complement pathway, especially with a focus on viral diseases, are still lacking [59].

## 5. Inborn Errors of Lymphocyte Function

Patients with inborn errors of lymphocyte function often present with a diverse spectrum of infections. 

The most extreme syndrome of lymphocyte dysfunction, severe combined immunodeficiency (SCID), arises from mutations in a number of genes: *IL2RG*, *RAG1*, *RAG2*, *ADA*, *JAK3*, and *IL7RG*. The patients are highly susceptible to lethal infections with a broad range of pathogens. HSV will usually cause recurrent skin infections and meningitis [104]. GATA2 deficiency mainly affects B cells, NK cells, and dendritic cells. The patients will also be at risk of disseminated HSV skin infections and meningitis, as well as hemophagocytic lymphohistiocytosis [104]. Intriguingly, HSE is not commonly found in SCID- or GATA2-deficient patients [104].

Errors in NK cell-mediated immunity show a well-established link to severe manifestations of HSV [105,106]. Of note, the original case of NK cell deficiency was later shown to have GATA2 deficiency [107]. However, reports of HSE in NK cell-deficient patients are few. Almerigogna and colleagues reported a series of five pediatric cases with HSE and impaired NK cell function from 2003 to 2005, two of which had a polymorphism in *CD16A* (p.Leu48His), although further functional analyses could not be carried out [108]. In a more recent report, Lisco and colleagues reported a male patient with a hemizygous germline variant in *IL2RG* (located on the X chromosome), which primarily affected his NK cells due to somatic reversion in the other lymphocyte populations [109]. The patient had a long history of severe and refractory human papillomavirus disease but notably also experienced HSE as an adolescent [109]. 

Common variable immunodeficiency (CVID) is an inborn error of B lymphocyte maturation and the most common immunodeficiency in adults. It is defined by low amounts of IgG and IgA or IgM, corresponding to a lack of proper seroconversion and insufficient response to polysaccharide vaccines without other causes for hypogammaglobulinemia [110]. Only a few cases have linked CVID to HSE or other severe manifestation of HSV infection, and the association remains weak [95,111].

## 6. Other Severe Phenotypes of HSV Infection

Extreme phenotypes of HSV infection other than HSE have been ascribed a genetic component, albeit with less systematic functional validation, but further studies may prove fruitful in understanding the host–pathogen interaction of HSV. 

Neonatal HSV infections can present as skin, eye, and mouth (SEM) infections or a disseminated phenotype involving hepatosplenomegaly. A recent genetic study of 10 infants with severe neonatal HSV (SEM, HSE, or disseminated disease) identified variants in genes known from HSE immunogenetics, including variants in *TLR3*, *TRAF3*, *IRF3*, *STAT1*, and *DBR1*, as well as candidate variants in other genes, including *MSR1*, *PRF1*, *GRB2*, *RAG2*, *C6*, and *C7* [112]. However, functional consequences of dominant disease-causing effects of any of these variants remain to be established.

Eczema herpeticum (EH) is a rare skin manifestation of HSV infection that can afflict patients with skin conditions, especially atopic dermatitis. Compromised epidermal barrier function is central to the pathogenesis and facilitates viral penetration deep into the skin [113]. Genetic studies have mainly focused on the general pathogenesis of atopic dermatitis, and there are only a few reports associating genetic variants with EH [113]. In 49 patients with recurrent EH, variants were found in eight genes: *RBBP8NL*, *TRIM15*, *CLEC7A*, *SIDT2*, *FRMD3*, *TPSG1*, *GSTZ1*, and *SP110* [114]. Of the identified variants, *RBBP8NL* and *SIDT2* could be linked to enhanced viral replication, restricted IFN-κ expression (an IFN-I originally named after its high expression in keratinocytes [115]), and the expression of genes related to keratinocyte differentiation [114]. While *SIDT2* is required for IL-1β expression during HSV-1 infection, *RBBP8NL* appears to restrict the expression of IL-1β. In addition to IFN-κ and IL-1β, IFN-γ plays an important role in protection against EH. Indeed, targeted deep sequencing in 121 EH patients revealed rare missense variants in the IFN-γ receptor 1 (IFNGR1), leading to decreased STAT1 signaling upon IFN-γ stimulation [116]. Finally, the importance of impaired barrier function in EH pathogenesis is reflected by loss-of-function variants in Filaggrin (*FLG*), detected in about 25% of EH patients and rarely in healthy individuals [117]. Filaggrin is important for the structural integrity of keratin filaments, while its degradation products, urocanic acid and pyrrolidone carboxylic acid, contribute to natural skin moisturization and acidification [118]. The impaired restriction of HSV-1 observed in *FLG* knockdown in human keratinocytes has been suggested to be due to reduced acidification [119].

Hemophagocytic lymphohistiocytosis (HLH) is a hyperinflammatory, life-threatening disorder characterized by persistent fever, splenomegaly with cytopenia, hypertriglyceridemia, and hypofibrinogenemia and the presence of hemophagocytosis in the bone marrow [120]. Histiocytes exert hemophagocytic activity in the reticulo-endothelial system, which can be triggered by congenital defects in T and NK cell immunity (primary HLH) or secondary to infections, rheumatic diseases, and cancer [121]. Often, secondary HLH is triggered by the unchecked replication of viruses, of which the Epstein–Barr virus (EBV) is the most frequent and may be partly explained by mutations in genes relevant to EBV immunity [121]. HLH triggered by HSV-1 is extremely rare in adults but less so in neonates [120]. The literature on the genetic causes of HSV HLH is limited. While some clinical case reports of HSV-1 HLH [122,123,124,125,126] and HSV-2 HLH exist [127], new genetic variants have been discovered in only three studies to date: in a 19-year-old woman suffering from postpartum HSV-1 HLH, a Caspase-8 (*CASP8*) variant was noted, but a definitive link between the variant and the impaired IFN responses observed in her PBMCs was not established [128]; a novel variant in *EP300* was discovered in a patient with Rubinstein–Taybi syndrome and HSV-1 HLH [129]; and finally, an 18-year-old female patient was reported in whom a *GATA2* variant was identified postmortem, although the disease-causing potential was not functionally validated [130].

## 7. Perspectives and Future Directions

CNS infections with HSV are extremely rare events caused by ubiquitous pathogens. These and other very rare infectious diseases have, therefore, been suggested as manifestations of novel IEI [131]. Using patient phenotypes as a steppingstone and harnessing the power of next-generation sequencing have proven immensely successful in defining novel syndromes of immunodeficiency with both broad and specific susceptibility to severe or lethal infections [132,133]. After the first discoveries of *STAT1* and *UNC93B1* deficiencies as causal of HSE, the focus has mainly been on unraveling other defects in the IFN-I system (Figure 1 and Table 1). Perhaps surprisingly, because HSV is a DNA virus, almost all defects in IFN-I activation have been detected in proteins related to the RNA-sensing TLR3 and RIG-I pathways. It is unlikely only to be due to confirmation bias that researchers focus on the pathways we know are crucial for HSV immunity, as important DNA-detecting pathways, such as cGAS-STING and TLR9, have been known for years, and predicted deleterious variants in genes related to these would have prompted further investigations. Rather, it may simply be that some of these pathways are not only non-redundant in human immunity but also more broadly in homeostasis, applying a strong negative selection on deleterious variants and thus a survivorship bias. Additionally, applying findings in mouse models directly to human immunology can easily mislead [134]. However, the argument of survivorship bias could also have been made for IFN-I signaling defects in general because of their key roles in antiviral immunity, but not only have many different deleterious variants been described, recently, it has also been shown that deleterious variants in *IFNAR1* and *IFNAR2* are prevalent at an appreciable rate in Western Polynesian and Arctic peoples [89,135]. Studies of these populations could, for the first time, provide insights into the clinical penetrance of severe infections in individuals without functional IFN-I signaling pathways, including HSV CNS infections.

Although neurons may not be the main source of IFN-I in the CNS, lesions of the TLR3 pathway, nevertheless, lead to impaired viral control, predisposing to HSE. Interestingly, these patients do not suffer from disseminated mucocutaneous viral disease in the periphery, as is the case for patients with general lymphocyte defects [104]. One way to reconcile these observations, is the hypothesis that TLR3 is a rheostat of tonic IFN-I signaling in the CNS. Tonic IFNAR signaling is well described in mice, with implications for hematopoietic stem cells, immune functions, NK cell proliferation, and bone remodeling, but the regulation of this basal level of IFN-I has not been fully elucidated [136]. Indeed, it was recently shown that patients with lesions of the TLR3-signaling pathway had significantly decreased levels of basal IFN-β as well as several ISGs [85]. The authors posit that the lack of tonic IFNAR signaling in TLR3- and IFNAR-signaling-deficient patients leads to unchecked early viral replication, predisposing to the development of HSE. It is, however, not yet clear what ligand, if any, maintains this constitutive TLR3 activation in healthy individuals [85] (Figure 3). 

In recent years, other parts of the immune system, such as cell autonomous innate immune mechanisms, have been implicated as non-redundant in anti-HSV immunity, including autophagy, small nucleolar RNAs, and RNA lariat metabolism (Figure 4). A large body of literature already exists on how autophagy works, but specific knowledge on which role it plays in anti-HSV immunity in humans, including differences between HSV-1 and HSV-2, is still needed. Autophagy also regulates the levels of many proteins, including several in the IFN-I system [137], blurring the lines between strictly autophagy-mediated immunity and IFN-I. Autophagy defects have, however, already been implicated not only in Mollaret’s meningitis but also in poliomyelitis, implying a non-redundant role in neuronal protection against specific neurotropic viruses [83,138]. 

Advances in our basic understanding of immunology is crucial to the discovery of novel immunodeficiencies. Without the discovery of *RNA5SP141*-mediated RIG-I activation, the connection between *GTF3A* and HSE is unlikely to have been made [33,75]. However, the opposite is also true, as evidenced by the body of literature, testifying to the ability of the discoveries of novel IEI in humans to teach us as yet uncovered principles of antiviral immunity in humans. With the characterization of variants in *DBR1* and *SNORA31* [79,81], translational immunology not only identifies which genes are central to immunity against which pathogens in natura but also provides basic immunology and virology with a challenge to understand novel mechanisms of neuroprotection in humans. Based on recent developments in the investigation of IEI in HSV immunity, it seems plausible that we will learn of new basic cellular processes involved in antiviral defenses and promoting severe disease and CNS infection when defective or dysregulated. Such pathways may include DNA sensing, cell death pathways, inflammasome activation, integrated stress responses, ubiquitination, and more. A guiding principle seems to be the quest to balance powerful antiviral neuroprotective immune responses aimed at eradicating the viral threat, while at the same time limiting inflammation and neuropathology and restoring homeostasis [96,139]. It will be exciting to follow the next technological advances and major discoveries, which may provide insights into the complexity of regulatory signaling networks in the immune system, and along this path, allow the discovery of novel IEI, the understanding of which may help diagnose patients and improve prophylaxis and treatment of severe herpesvirus infections of the CNS.

## Figures and Tables

**Figure 1 pathogens-12-00310-f001:**
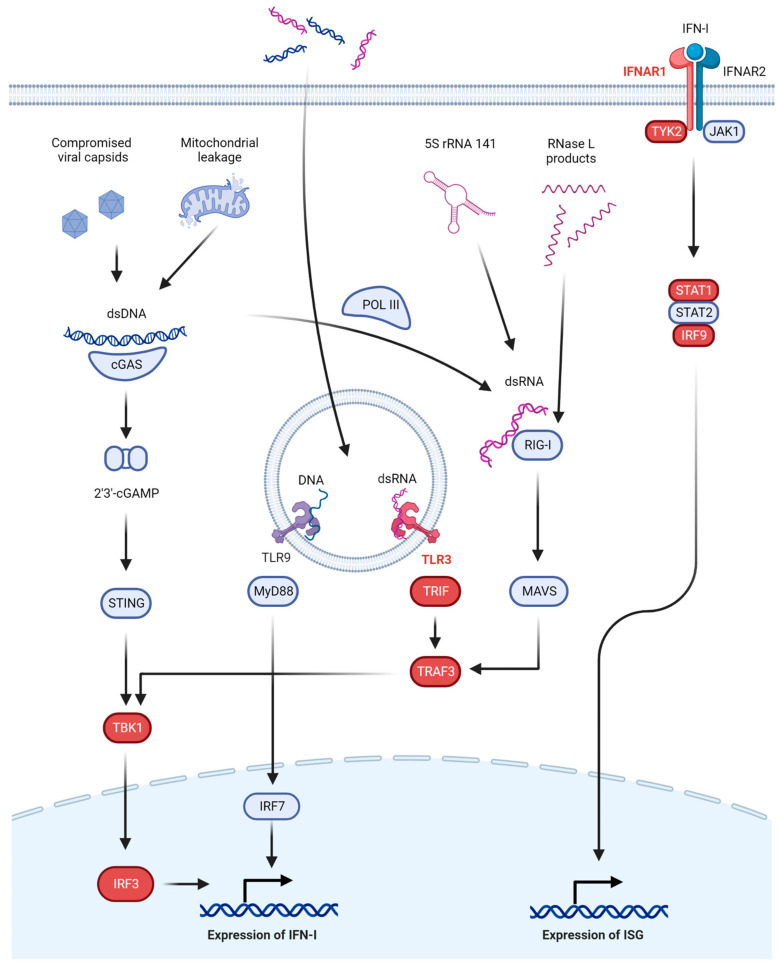
Interferon-dependent antiviral immunity to herpes simplex virus. Major pathways involved in innate anti-HSV immunity based on IFN-I are shown. In the endosomal compartment, dsRNA is recognized by TLR3 and dsDNA by TLR9. In the cytosol, dsDNA is recognized by cGAS to produce 2′3′-cGAMP and activate the STING pathway or by POL III to produce RNA. dsRNA in the cytosol is recognized by RIG-I to activate the MAVS pathway. These pathways (except for TLR9) converge at TBK1, finally phosphorylating IRF3 and activating IFN-I production. IFN-I acts through its receptor IFNAR to activate ISGs and drive cellular antiviral programs. Proteins in which defects have been linked to HSE are shown in red.

**Figure 2 pathogens-12-00310-f002:**
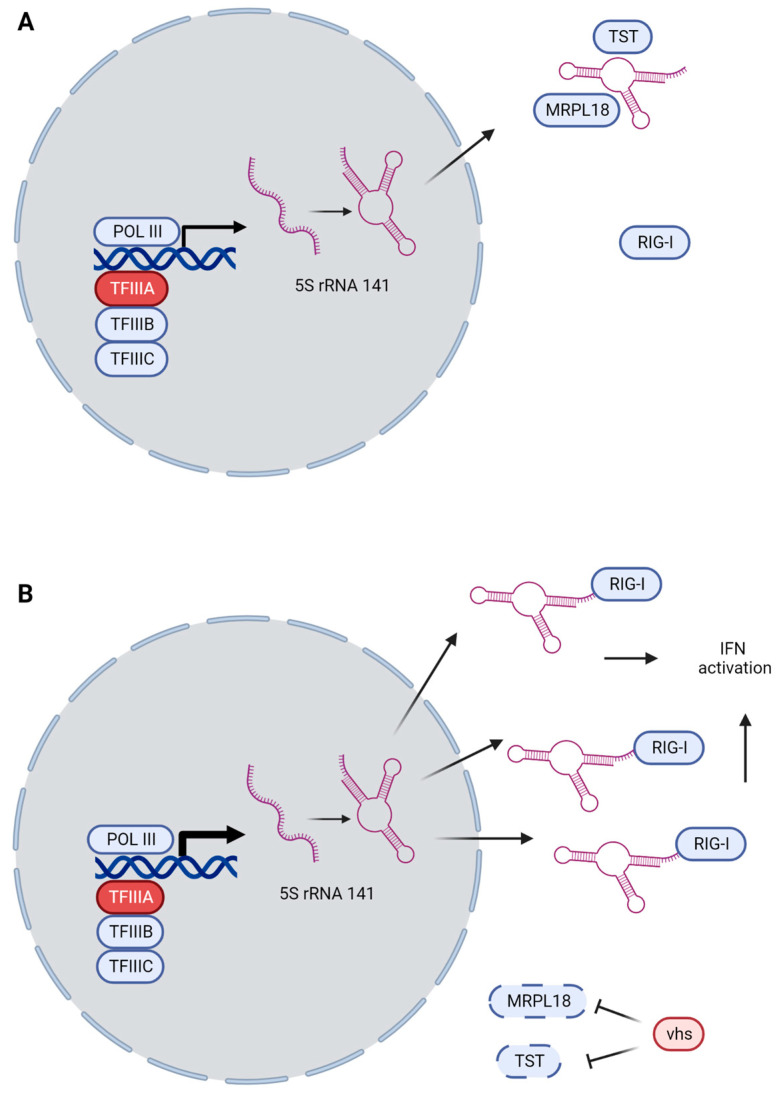
Guard mechanism of *RNA5SP141* and RIG-I. (**A**) During homeostasis, POL III transcribes *RNA5SP141* at low levels, with the product staying mostly in the nucleus. When escaping to the cytosol, the RNA is sequestered by MRPL18 and TST, preventing it from activating RIG-I. (**B**) Upon HSV-infection, vhs causes global mRNA downregulation, decreasing the amounts of MRPL18 and TST. *RNA5SP141* is slightly upregulated, but importantly, the transcript is shifted from the nucleus to the cytoplasm where it, disinhibited by the lack of MRPL18 and TST, can activate RIG-I and induce IFN-dependent immunity. TFIIIA (red, encoded by *GTF3A*) is part of the transcription factor complex necessary for POL III-transcription of 5S rRNA species.

**Figure 3 pathogens-12-00310-f003:**
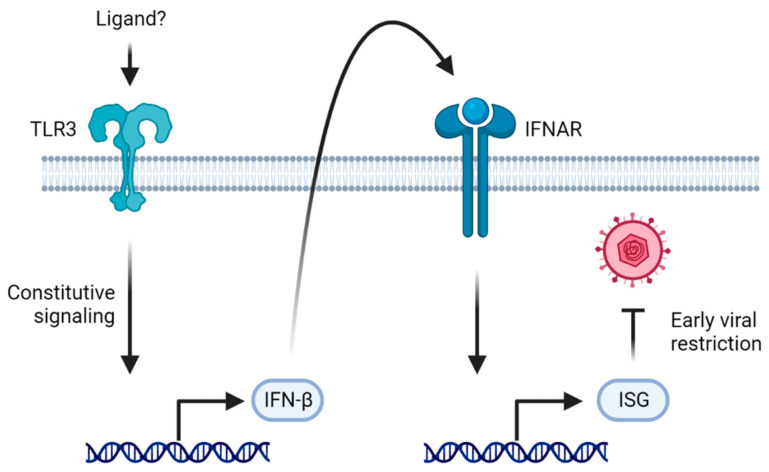
TLR3 as a rheostat of tonic IFNAR signaling. In healthy individuals, TLR3 signals constitutively, leading to a basal level of IFN-β which activates IFNAR signaling. The resulting basal level of ISGs being expressed restricts early viral replication. In patients deficient in the TLR3-IFNAR-signaling axis, the virus replicates at a higher rate early during the infection, predisposing to HSE.

**Figure 4 pathogens-12-00310-f004:**
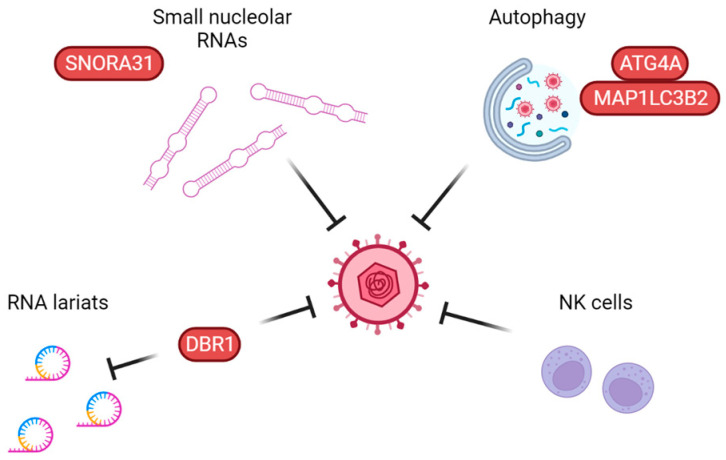
Proposed non-redundant anti-herpes simplex virus pathways working independently of type I interferons. Small nucleolar RNAs, including snoRNA31, have been shown to exert a direct antiviral effect on HSV replication. Autophagy defects have been linked to two adult cases of Mollaret’s meningitis. NK cell deficiency has been identified in a few cases of pediatric HSE. Finally, DBR1 debranches RNA lariats and restricts HSV replication, although the exact antiviral mechanism is not yet fully understood. Proteins in which defects have been linked to HSE are shown in red.

**Table 1 pathogens-12-00310-t001:** List of genetic variants in innate immune pathways associated with heightened susceptibility to HSV CNS infections. In some cases, several variants have been shown to cause deficiency of a specific protein; here, variants from patients in whom HSV-1 or HSV-2 CNS infection was diagnosed are listed.

Disease	Signaling Pathway Affected	Gene	Variants	References
HSE	TLR3	*UNC93B1*	c.1034del4	[64]
			c.781G>A	
		*TLR3*	p.P554S	[65,66,67]
			p.E746X	
			p.G743D	
			p.R811I	
			p.D592N	
			p.M374T	
			p.L360P	
			p.R867Q	
		*TRIF*	p.R141X	[68]
			p.S186L	
	TLR3/RIG-I	*TRAF3*	p. R118W	[69]
	TLR3/RIG-I/STING	*TBK1*	p.G159A	[70]
			p.D50A	
		*NEMO*	p.M38fsX48	[71,72,73]
		*IRF3*	p.R285Q	[74]
	5S rRNA 141/RIG-I	*GTF3A*	p.C195W	[75]
			p.C219R	
	IFNAR signaling	*IFNAR1*	g.34,726,420_34,728,094del	[76]
		*STAT1*	c.1757–1758delAG	[77]
		*IRF9*	c.577+1G>T	[78]
	Small nucleolar RNA	*SNORA31*	n.36T>C	[79]
			n.75C>G	
			n.96T>G	
			n.111T>C	
	Lectin pathway *	*MASP2*	p.R203W	[80]
			p.G634R	
Brainstem encephalitis	RNA lariat metabolism	*DBR1*	p.I120T	[81]
Meningitis	IFNAR signaling	*TYK2*	p.R638X	[82]
Mollaret’s meningitis	Autophagy *	*ATG4A*	p.L90I	[83]
		*MAP1LC3B2*	p.L109M	[83]

* All variants listed have been identified in pediatric patients except for two adult cases of *MASP2* deficiency in HSE and two cases of autophagy defects in Mollaret’s meningitis.

## Data Availability

Not applicable.

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
