# Peer review of "Inborn Errors of Immunity Predisposing to Herpes Simplex Virus Infections of the Central Nervous System"

_pathogens, 2023, doi:10.3390/pathogens12020310_

Round 1

Reviewer 1 Report

In this manuscript, Skouboe et al. present a review on the subject of inborn errors of immunity (IEI) that predispose individuals to herpes simplex virus (HSV) infections of the central nervous system (CNS).  It begins with a succinct outline of the different types of diseases associated with HSV infection of the CNS, then provides an overview of different types of immune responses to HSV infection. The last two thirds of the review describe data linking mutations associated with various types of HSV-related immunity (interferon-dependent and -independent innate immunity, different types of cell intrinsic immunity) to patients with HSV infections of the CNS.  The subject of the review is important because HSV infections of the CNS are difficult to treat effectively even with antivirals such as acyclovir, and questions remain regarding the underlying mechanisms.  The review covers the essential major topics of this subject.  Although the last author was also an author on a publication in 2020 focused on IEI related to herpesvirus infections in general (Jouanguy et al. 2020), they do cite this paper, and the table listing genes associated with HSV CNS infections has been updated for this present manuscript.  It should also be mentioned that the last author published a paper less than a year ago on the same subject as this manuscript but that was written in Danish.  The weakest parts of the manuscript are the figures, which should be improved along with the corresponding legends.  Other detailed comments are listed below.

1. Line 77:  The number of viral proteins encoded by HSV-1 should be updated—there are many more than 70—and an appropriate reference should be used.

2. Fig. 1:  The figure should be refined.  One example: It is unclear from the picture what the relationship is between TBK1, IKKe and IRF3.  Also, the legend should be expanded to adequately describe the figure.

3. Table 1:  In the text, the authors point out the two age blocks associated with HSV CNS disease.  It would be helpful if the ages of the patients were indicated in the table where possible.  (At a minimum, pediatric and adult cases should be differentiated.)

4. Fig. 2:  Minor point, but the endonuclease vhs is released from the viral tegument.  Therefore, the figure of the HSV particle shown that includes the envelope is not accurate because by this point the viral envelope has fused with the cellular membrane.

5. Line 381: The authors should explicitly state the function of Filaggrin so that the readers can understand the conclusions drawn regarding mutations in the FLG gene.

6. Fig. 4: It is a good choice to include a figure specifically showing interferon-independent anti-HSV pathways; however, this figure should be improved.  The figure should be modified to better illustrate the concepts that are presented, and the legend must be expanded. (Two examples: [1] It is unclear what the double arrow means.  [2] The figure appears to be indicating that NK cell deficiency blocks HSV infection).

Author Response

  1. Line 77: The number of viral proteins encoded by HSV-1 should be updated—there are many more than 70—and an appropriate reference should be used.

Re: We agree with the reviewer and have adjusted the text accordingly, lines 78-79. A new reference (Whisnant et al, 2020) has been added.

  1. Fig. 1: The figure should be refined. One example: It is unclear from the picture what the relationship is between TBK1, IKKe and IRF3.  Also, the legend should be expanded to adequately describe the figure.

Re: The complex interplay between TBK1 and IKKe is, indeed, outside the scope of this review, since no defects have been described, connecting IKKe with HSV CNS infections. Accordingly, we have decided to exclude IKKe from the figure. We have also expanded the legend as suggested.

  1. Table 1: In the text, the authors point out the two age blocks associated with HSV CNS disease. It would be helpful if the ages of the patients were indicated in the table where possible.  (At a minimum, pediatric and adult cases should be differentiated.)

Re: We agree with the reviewer that this distinction is important and have therefore added a note to the table specifying which cases are adult and which are pediatric.

  1. Fig. 2: Minor point, but the endonuclease vhs is released from the viral tegument. Therefore, the figure of the HSV particle shown that includes the envelope is not accurate because by this point the viral envelope has fused with the cellular membrane.

Re: To avoid confusion, the HSV virion has been removed from the figure.

  1. Line 381: The authors should explicitly state the function of Filaggrin so that the readers can understand the conclusions drawn regarding mutations in the FLG gene.

Re: We agree, and to clarify we have added a brief description of the functions of Filaggrin as suggested, lines 390-394.

  1. Fig. 4: It is a good choice to include a figure specifically showing interferon-independent anti-HSV pathways; however, this figure should be improved. The figure should be modified to better illustrate the concepts that are presented, and the legend must be expanded. (Two examples: [1] It is unclear what the double arrow means. [2] The figure appears to be indicating that NK cell deficiency blocks HSV infection).

Re: We thank the reviewer for pointing out that this figure needs clarification and have changed it accordingly. Regarding DBR1, we have inserted the protein between RNA lariats and the virus, as it cannot be deduced from the paper whether the DBR1 restricts HSV through debranching of RNA lariats or if it has direct antiviral functions. Regarding NK cell deficiencies, the text has been corrected. Furthermore, we have expanded the figure legend, lines 475-480.

Reviewer 2 Report

In the  manuscript entitled: "Inborn errors of immunity predisposing to herpes simplex virus infections of the central nervous system" Skouboe et al. in a very systematic manner review published data related to inborn errors of both IFN-dependent and IFN-independent mechanisms of immunity to HSV infections in the CNS.

The manuscript is well written and organized giving a comprehensive synopsis of the addressed topic. The introduction gives a very good outline of the HSV infection mechanism in the CNS and points out the main players in the immune response to the infection. It is accompanied by a very clear and helpful figure representing overview of interferon-dependent antiviral immunity to HSV. The main part of the review gives a very thorough summary of studies relating to inborn errors of immunity to HSV, again devided into interferon-dependent and -independent categories. The data is supported by a table listing the genetic variants involved in innate immunity associated with higher sensitivity to HSV CNS infections.

I have only minor poins/suggestions:

1. I find that it would be helpful to the reader to be referred to Fig. 1 already at the beginning of the section 2.1.

2. It might be beneficial to include, wherever possible, points of interference of the virus/viral proteins with the pathways depicted in Fig.1.

3. For table 1 would it be possible to put the frequency of the listed variants in the population or otherwise compare the frequency of the variants with the frequency of the HSV CNS infections?

Author Response

  1. I find that it would be helpful to the reader to be referred to Fig. 1 already at the beginning of the section 2.1.

Re: A reference to Figure 1 has been included line 85.

  1. It might be beneficial to include, wherever possible, points of interference of the virus/viral proteins with the pathways depicted in Fig.1.

Re: Although informative, we believe that adding interactions between HSV and host factors, which is a complex subject with multiple known interactions, would unnecessarily complicate the figure, distracting from the main host pathways that we want to highlight and which are the main topic of the review.

  1. For table 1 would it be possible to put the frequency of the listed variants in the population or otherwise compare the frequency of the variants with the frequency of the HSV CNS infections?

Re: Most variants described in the literature are, so far, very rare or even private to the respective kindreds. We therefore do not believe that there is much added value in the frequency of each variant, but rather we want to show that deleterious variants are clustering in specific pathways. We have added the estimated frequency of HSE of 1-2/250,000 per year line 48.

We thank the reviewer for the positive evaluation of our manuscript